# Performance Evaluation of STANDARD Q COVID/FLU Ag Combo for Detection of SARS-CoV-2 and Influenza A/B

**DOI:** 10.3390/diagnostics13010032

**Published:** 2022-12-22

**Authors:** Kristin Widyasari, Soomin Kim, Sunjoo Kim, Chae Seung Lim

**Affiliations:** 1Department of Laboratory Medicine, Gyeongsang National University Changwon Hospital, Changwon 51472, Republic of Korea; 2Department of Laboratory Medicine, Institute of Health Sciences, Gyeongsang National University College of Medicine, Jinju 52727, Republic of Korea; 3Department of Laboratory Medicine, College of Medicine, Korea University, Seoul 08307, Republic of Korea

**Keywords:** rapid antigen test, SARS-CoV-2, influenza A, influenza B, point-of-care

## Abstract

We evaluated the performance of the STANDARD Q COVID/FLU Ag Combo test (Q Ag combo test) for the detection of SARS-CoV-2, influenza A, and influenza B using a single point-of-care device compared with real-time PCR. A total of 408 individuals, 55 positives with SARS-CoV-2, 90 with influenza A, 68 with influenza B, and 195 negatives for all viruses, participated. The Q Ag combo test demonstrated a high level of sensitivity of 92.73% and a specificity of 99.49% for the detection of SARS-CoV-2. When the number of days from symptom onset (DSO) was restricted to 0 < DSO ≤ 6, the sensitivity of the Q Ag combo test to detect SARS-CoV-2 was 100%, and when the Ct value of *RdRp* was ≤20, the sensitivity to detect SARS-CoV-2 was 93.10%. The Q Ag combo test results also demonstrated a sensitivity of 92.22% and a specificity of 100% for influenza A, a sensitivity of 91.18%, and a specificity of 99.49% for influenza B. The agreement analysis of the Q Ag combo test with the RT-PCR results demonstrated excellent outcomes, making it useful and efficient for the detection of SARS-CoV-2, influenza A, and influenza B.

## 1. Introduction

Respiratory tract infections (RTIs) are the most common and challenging clinical diseases and have high rates of morbidity and mortality worldwide. RTIs are usually grouped into upper and lower respiratory infections, with viruses responsible for about 90% of upper respiratory infections [1,2]. Among the numerous RTIs that can be named, there are four main viral RTIs that are commonly found nowadays: COVID-19, influenza, respiratory syncytial virus (RSV) infection, and the common cold [3]. Simultaneous infections of the respiratory tract by multiple viruses are common in hospitalized patients. Even though there is no evidence of whether co-infections cause more severe clinical outcomes than infections with a single virus, the rapid and accurate detection and identification of viruses in a patient’s specimens is crucial for clinical diagnosis, isolation, and appropriate treatment, as well as to prevent further spread of the diseases [4].

COVID-19 is an infectious disease caused by the SARS-CoV-2 virus. COVID-19 was first identified in Wuhan, China at the end of 2019, and on 11 March 2020, it was announced as a pandemic by the World Health Organization [5]. Despite multiple efforts, including the global administration of COVID-19 vaccines, COVID-19 infection cases keep resurging. In Korea, a cumulative ten million confirmed COVID-19 cases were recorded in the first two years after the disease emerged, whereas this number keeps increasing significantly with up to twenty million cases being recorded in a period of only 133 days [6].

Influenza is also a highly contagious disease caused by influenza viruses, such as influenza A, B, and C viruses. Influenza causes different degrees of illness, from mild to severe, and can sometimes lead to mortality [7,8]. Similar to COVID-19, the major control measure for the prevention and control of influenza infections is the administration of vaccines. Nevertheless, the high mutation rate of influenza viruses allows them to evade vaccine-induced immune responses [7]. Each year, influenza places a substantial burden on the health of people worldwide. An outbreak of influenza occurs every year, mostly during the winter season in both the northern and southern hemispheres and year-round in tropical areas [9].

Both SARS-CoV-2 and influenza viruses are highly transmissible through direct contact with an infected person or through droplets made when infected individuals cough, sneeze, or talk [10,11]. COVID-19 and influenza also share some similarities in terms of symptom development, including cough, sore throat, fever, and headache, and sometimes can be fatal in the elderly [12]. Thus, rapid and accurate differential diagnosis is a crucial first step to preventing the further spread of the disease and providing proper treatment.

In previous years, numerous antigen-specific point-of-care testing kits have been developed and implemented for the detection of either COVID-19 or influenza [13,14,15]. However, most of the currently available point-of-care testing kits are designed to specifically identify a single virus. A testing kit that enables the detection of multiple viruses from a single specimen using a single device would be favorable for use in rapid on-site testing as this would significantly decrease the test turnaround time. In this study, we evaluated the performance of the STANDARD Q COVID/FLU Ag combo, a rapid antigen detection kit that enables the detection of SARS-CoV-2, influenza A, and influenza B from the nasopharyngeal specimens of patients with suspected respiratory infections. We also evaluated the agreement of the STANDARD Q COVID/FLU Ag Combo with real-time PCR (RT-PCR) results as the golden standard of assessment.

## 2. Materials and Methods

### 2.1. Subjects

A total of 408 samples were evaluated in this study. All the samples were assessed by real-time PCR for detection of SARS-CoV-2, and the influenza A, and influenza B viruses. Furthermore, these samples were assigned to four groups: positive SARS-CoV-2 (55 samples), positive influenza A (90 samples), positive influenza B (68 samples), and negative for all tested viruses (195 samples) (Figure 1). Each SARS-CoV-2-infected patient who visited the Gyeongsang National University Changwon Hospital (GNUCH) between May and August 2022 participated in this study prospectively. These patients were either asymptomatic or mildly symptomatic for respiratory illnesses. Influenza A positive samples used in this study were frozen-stored samples from the Korea University Guro Hospital (KUGH) that were collected during the last 4 years. All positive samples of influenza B and a portion of the influenza A samples were purchased from overseas biobanks (National Institute of Hygiene and Epidemiology, Hanoi, Vietnam; and Instituto de Investigación Nutricional, Lima, Peru). Uninfected controls were randomly selected from the patients who were admitted to GNUCH for the management of other respiratory diseases during the same period and were screened for COVID-19, influenza A, or influenza B by real-time PCR before or during admission. Individuals under 19 years of age were excluded from this study. Participants who were COVID-19 positive or negative agreed to this study and submitted their written informed consent.

### 2.2. Sample Collection

The nasopharyngeal samples for assessment of COVID-19 and negative control were collected as described previously [13]. In short, a flocked swab (NFS, Noble Biosciences, Hwasung, Korea) was gently inserted through the nostril to a depth of 5 to 7 cm parallel to the palate and gently rubbed and rolled in place for several seconds to absorb secretion. Subsequently, the swab was gently removed, and the tip of the swab was placed in the buffer tube before being subjected to analysis using the STANDARD Q COVID/FLU Ag Combo test (SD BIOSENSOR, Suwon, Korea).

### 2.3. The Rapid Combo Antigen Test (RCAT)

The STANDARD Q COVID/FLU Ag Combo test is a rapid chromatographic immunoassay for the qualitative detection of specific SARS-CoV-2, influenza A, and influenza B antigens present in human nasopharyngeal specimens using a single device [16]. The STANDARD Q COVID/FLU Ag Combo test hereinafter referred to as the Q Ag combo test was performed by the manufacturer’s instructions. Briefly, following sample collection, a swab in the buffer tube was moved around in a circle in place and squeezed into the tube wall. Subsequently, three drops of the reaction mixture were applied to the device. The results appeared as a band(s) of color after 15 min and were interpreted (Figure 2).

The test was conducted by three technicians who were blinded to avoid any bias from the observer. The results were interpreted as negative when only one band appeared on the C (control) line. The results were interpreted as positive when bands appeared on both the C and the tested lines (S, SARS-CoV-2; A, influenza A; and B, influenza B) (Figure 3), and as invalid if no band appeared or if the bands only appeared on the tested lines but not on the C line.

### 2.4. The Rapid Antigen Test (RAT)

The agreement of the Q Ag combo test kit in detecting SARS-CoV-2, influenza A, and influenza B was evaluated with the pre-existing RATs that specifically detect a single virus. The Panbio COVID-19 Ag Rapid Test Device (Abbot Korea, Yongin, Korea) and the SD Bioline Influenza Ultra Rapid Test kit (Abbot Korea) were used for this purpose. The RATs were performed by the manufacturer’s instructions with small modifications due to the availability of samples for the influenza A/B test (frozen–thawed nasopharyngeal samples were used for the influenza A/B test). The results appeared as a band(s) of color after 15 min and were interpreted as described above.

### 2.5. Real-Time Reverse Transcription-Polymerase Chain Reaction (RT-PCR)

Before being subjected to testing using the RACT/RATs, all samples were confirmed to be negative or positive for SARS-CoV-2, influenza A, and influenza B by RT-PCR. The Allplex™ SARS-CoV-2/FluA/FluB/RSV Assay (Seegene, Seoul, Korea), a multiplex RT-PCR assay used for the simultaneous detection of SARS-CoV-2, influenza A/B, and RSV, was used for this confirmation. The RT-PCR assay targets an RNA-dependent RNA polymerase (*RdRp*), spike (S) and nucleocapsid (N) genes of SARS-CoV-2, and specific genes of influenza A, influenza B, and RSV. The results were interpreted as positive only if the cycle threshold (Ct) value is within the cutoff values and negative if they are outside the cutoff or if there is no amplification. According to the manufacturer’s instructions, the cutoff values are Ct ≤ 40 for *RdRp* and Ct ≤ 38 for *S*, *N*, influenza A, and influenza B specific genes.

### 2.6. Statistical Analysis

All tests were performed in a blinded manner. The diagnostic performance of the Q Ag combo test, including its sensitivity, specificity, positive predictive value, and negative predictive value was determined using a comparison analysis against results from the RT-PCR as the gold standard. The agreement between the Q Ag combo test and pre-existing rapid Ag tests was assessed based on the kappa index. We also performed sensitivity analyses by restricting COVID-19-positive cases according to the number of days from symptom onset (DSO) (0–3, 4–6, and >7) and the *RdRp* Ct values (Ct ≤ 20, 20 < Ct ≤ 30, 30 < Ct < 40). When no symptoms occurred, DSO was defined as the number of days from sample collection to RT-PCR confirmation. We performed all statistical analyses using SAS software version 9.4 (SAS Institute Inc., Cary, NC, USA).

## 3. Results

### 3.1. Agreement and Diagnostic Performance of STANDARD Q COVID/FLU Ag Combo Compared with RT-PCR for the Detection of SARS-CoV-2

The results for the agreement of the Q Ag combo test and RT-PCR in the detection of SARS-CoV-2 from the nasopharyngeal samples are presented in Table 1. Cohen’s kappa index value was 0.940, indicating substantial agreement between the two detection methods. Furthermore, the sensitivity of the Q Ag combo test when used to detect SARS-CoV-2 from the nasopharyngeal samples that had been confirmed to be positive by RT-PCR was 92.73% (95% confidence interval, CI, 82.41–97.98%). Correspondingly, the specificity of the Q Ag combo test was 99.49% (95% CI, 97.18–99.99%), while the positive and negative predicted values were 98.08% (95% CI, 87.82–99.72%) and 97.98% (95% CI, 94.97–99.20%), respectively (Table 1).

### 3.2. Agreement and Diagnostic Performance of the STANDARD Q COVID/FLU Ag Combo Compared to the RT-PCR for the Detection of Influenza A

The Q Ag combo test and RT-PCR also demonstrated strong agreement in terms of their ability to detect influenza A from the frozen–thawed nasopharyngeal samples with a Cohen’s kappa value of 0.941 (Table 2). The sensitivity of the Q Ag combo test to detect influenza A was as high as 92.22% (95% CI, 84.63–96.82%) with a specificity of up to 100% (95% CI, 98.13–100%). The positive predicted value was as high as 100% (95% CI, 95.65–100%) with a negative predicted value of 96.53% (95% CI, 93.19–98.27%) (Table 2).

### 3.3. Agreement and Diagnostic Performance of the STANDARD Q COVID/FLU Ag Combo Compared to RT-PCR for the Detection of Influenza B

The Q Ag combo test also demonstrated strong agreement with the RT-PCR when used to detect influenza B from the frozen–thawed nasopharyngeal samples. The Cohen’s kappa index of the Q Ag combo test results and the RT-PCR results were as high as 0.928 (Table 3). The performance evaluation demonstrated that the sensitivity level of the Q Ag combo test in detecting influenza B was 91.18% (95% CI, 81.78–96.69%) with a specificity of up to 99.49% (95% CI, 97.18–99.99%). The positive and negative predicted values of the Q Ag combo test for the detection of influenza B were 98.41% (95% CI, 89.76–99.77%) and 97% (95% CI, 93.77–98.58%), respectively (Table 3).

### 3.4. Diagnostic Performance of the STANDARD Q COVID/FLU Ag Combo for the Detection of SARS-CoV-2 by Restricting Positive Cases According to the DSO and Ct Values

To assess whether the ability of the Q Ag combo test to detect SARS-CoV-2 is affected by changes in certain conditions, we restricted COVID-19 positive cases according to the number of days after symptom onset (DSO) as well as restricting the cycle threshold value (Ct value) of *RdRp*. Our analysis demonstrated that the use of 0<DSO≤6 days resulted in a sensitivity level of 100% (95% CI, 84.56–100%). However, with DSO >7 days, the sensitivity level of the Q Ag combo test when detecting SARS-CoV-2 decreased to 63.64% (95% CI, 30.79–89.07%) (Table 4).

Furthermore, we performed a sensitivity analysis by classifying COVID-19-positive samples according to the Ct value of the *RdRp* (Table 4). We assessed the samples that were diagnosed as positive by RT-PCR, within which Ct value intervals were divided into Ct ≤ 20, 20 < Ct ≤ 30, and 30 < Ct ˂ 40. The sensitivity of the Q Ag combo test for samples with Ct ≤ 20 was 93.10% (95% CI, 77.23–99.15%). Subsequently, when we increased the interval of Ct values to between 20 and 30 (20 < Ct ≤ 30), the sensitivity of the Q Ag combo test declined slightly to 88.89% (95% CI, 65.29–98.62%). Lastly, among the samples with Ct values of 30 < Ct < 40, the sensitivity level of the Q Ag combo test was 100% (95% CI, 63.06–100%).

### 3.5. Agreement of the STANDARD Q COVID/FLU Ag Combo with the Pre-Existing Rapid Antigen Tests

We also evaluated the agreement of the Q Ag combo test with other pre-existing RAT kits. The Q Ag combo test demonstrated strong agreement with the Panbio COVID-19 kit for the detection of SARS-CoV-2 with a Cohen’s kappa value of 0.924 (Table 5). Correspondingly, the Q Ag combo test also demonstrated strong agreement with the SD Bioline Influenza Ultra Rapid test kit for the detection of influenza A with a Cohen’s kappa value of up to 0.983 (Table 5). However, the agreement between the Q Ag combo test and the SD Bioline Influenza Ultra Rapid test kit for the detection of influenza B was weak (Cohen’s kappa value = 0.299) (Table 5).

## 4. Discussion

COVID-19 and influenza are highly contagious and share many similarities in terms of symptoms and signs. Furthermore, there is also a high possibility of co-infection with two or more viruses at the same time. Thus, a rapid and accurate identification method that enables the detection and differentiation of SARS-CoV-2 and influenza viruses with a single device is favorable for diagnostic laboratories due to its short turnaround time and cost-effectiveness.

To verify the performance of the Q Ag combo test for the detection of SARS-CoV-2, influenza A, and influenza B from a single sample using a single device, we performed an agreement analysis of the Q Ag Combo test with RT-PCR and pre-existing rapid Ag test kits. We also performed a sensitivity analysis by assessing the samples that had been diagnosed as positive for SARS-CoV-2, influenza A, or influenza B by RT-PCR.

Our study demonstrated strong agreement between the Q Ag combo test and the RT-PCR for the detection of SARS-CoV-2, influenza A, and influenza B viruses with Cohen’s Kappa values that were considered to indicate an almost perfect agreement [17]. The sensitivity levels were higher than those previously reported for combo rapid antigen test kits. The sensitivity level of the rapid test kit for SARS-CoV-2 is reported to be up to 80.9% [18]. Moreover, the sensitivity and specificity levels of the Q Ag combo test in our study are higher than the criteria recommended by the US Food and Drug Administration (FDA), which states that the sensitivity and specificity of rapid antigen test kits should be at least 80% and 98%, respectively [19]. Hence, the ability of the Q Ag combo test used in our study to discriminate SARS-CoV-2, influenza A, and influenza B viruses from a single sample using a single device is regarded as exceptional.

Previously, we reported that the duration from symptom onset (DSO) and the cycle of threshold (Ct) values determine the sensitivity of rapid single antigen tests in terms of their ability to detect SARS-CoV-2 in the nasopharyngeal samples [13]. Therefore, in this study, we restricted the DSO and the Ct value of *RdRp* for SARS-CoV-2 and analyzed the sensitivity of the Q Ag combo test when it was used to detect the presence of SARS-CoV-2 in the samples according to the different DSOs and Ct values. The sensitivity level of the Q Ag combo test was as high as 100% when used to assess the samples collected within a week (0–6 days). However, when used to assess samples collected at DSO > 7 days, the sensitivity level of the Q Ag combo test decreased significantly. This result is consistent with earlier studies on the performance of rapid single antigen tests, which revealed that the kits’ sensitivity is optimal when used to evaluate samples obtained within a week (≤7 DSO) [13,20,21].

The Ct values in the RT-PCR represent the number of amplification cycles required for the target gene to exceed a threshold level [22]. The Ct values have been reported to be correlated with SARS-CoV-2 accumulation and the clinical conditions of patients; hence, they were assumed to be an appropriate surrogate for the viral load [23,24,25]. The sensitivity level of the Q Ag combo test when used to assess samples with Ct values of *RdRp* ≤ 20 was higher than that of samples with Ct values of 20 < Ct ≤ 30. Nonetheless, the sensitivity levels from both groups were higher than the sensitivity level recommended by the FDA [19], suggesting that the ability of the Q Ag combo test to detect SARS-CoV-2 in nasopharyngeal samples with Ct values ≤ 30 is outstanding. Previously, a study reported a poor positivity level for pre-existing rapid Ag tests when used to assess samples with Ct values above 30 [26]. Virus replication in samples with Ct values above 30 may be difficult and probably has no epidemiological relevance, so most of these samples produce negative results [27]. In contrast, we observed that all eight samples with Ct values of 30 < Ct < 40 in our study turned out to be positive when assessed using the Q Ag combo test.

To rationalize whether false positives occurred during this assessment, we compared these results with those obtained from the RT-PCR and confirmed that all eight samples were also classified as positive for COVID-19 by the RT-PCR. The median Ct value for these eight samples was 32.08 (cutoff value of *RdRp*’s Ct ≤ 40). Hence, the Q Ag combo test result is thought to be a true positive by the RT-PCR, which is used as the gold standard. However, we also considered the small number of samples included in this group (30 < Ct < 40) as the limitation that makes it difficult to determine whether the outcome of this group is a true finding, given that small sample size may reduce the statistical power.

We also compared the performance of the Q Ag combo test with that of pre-existing rapid antigen test kits for the detection of either SARS-CoV-2, influenza A, or influenza B. The Q Ag combo test demonstrated a strong agreement with the Panbio COVID-19 Ag rapid test kit when used to detect SARS-CoV-2. According to the manufacturer, the sensitivity level of the Panbio COVID-19 Ag rapid test kit is 98.1% and the specificity is 99.8% [28]. The strong agreement between the Q Ag combo test and the Panbio COVID-19 test suggests that the ability of the Q Ag combo test to detect SARS-CoV-2 in the samples is as good as that of the Panbio COVID-19 Ag rapid test kit. When used to detect influenza viruses, the Q Ag combo test demonstrated a strong agreement with the SD Bioline Influenza Ultra Rapid test kit for the detection of influenza A but showed a low level of agreement for the detection of influenza B. We found that among the 68 samples that had previously tested positive for influenza B by RT-PCR, only 14 samples were confirmed to be positive when re-assessed using the SD Bioline Influenza Ultra Rapid test kit. In contrast, when those 68 RT-PCR-confirmed samples were re-assessed using the Q Ag combo test, 62 samples were verified to be positive for influenza B.

The SD Bioline Influenza Ultra Rapid test kit manufacturer stated that the sensitivity and specificity of the kit are up to 91.5% and 98.7% for samples from the nasopharyngeal swab, and up to 91.7% and 98.9% for samples from the nasopharyngeal aspirate, respectively [29]. However, there is no information about the sensitivity and specificity of the SD Bioline Influenza Ultra Rapid test kit for assessment of the samples that have undergone freeze–thawed cycles. Note that the influenza B samples for our assessment were frozen stock samples. Previously, a manufacturer-independent evaluation of the Ag-rapid detection kit’s (Ag-RDT) limit of detection reported that 11 among 19 Ag-RDTs showed decreased sensitivity up to 20 folds when used to assess frozen-stock samples (−80 °C) [30]. Most of the RATs are intended to be point-of-care (POC) tests, hence requiring fresh samples to meet the highest sensitivity and specificity as stated by the manufacturer. Therefore, we assumed that repeated freeze–thawed cycles affected the sensitivity of the SD Bioline Influenza Ultra Rapid test kit (preexisting RAT), but not the Q Ag combo test kit, resulting in the discrepancy and a low Cohen’s Kappa value for agreement between the two test kits.

The use of frozen-stored samples, which leads to the discrepancy of the pre-existing RAT and Q Ag combo test for the detection of influenza B may be the biggest limitation in our study. The frozen-stored samples for the influenza A/B test were used in this study due to our inability to recruit participants with influenza A/B infections. As reported before, during the second winter season of the COVID-19 era (2021–2022 winter season), there was no influenza outbreak in South Korea [31]. Nevertheless, the Q Ag combo tests demonstrated a high sensitivity when used for the assessment of freshly collected or frozen-stored samples, making it a promising POC test kit with high performance for the detection of SARS-CoV-2, influenza A, and influenza B with only a single device, regardless of the sample condition.

## 5. Conclusions

In conclusion, the Q Ag combo test showed excellent sensitivity and specificity for the detection of SARS-CoV-2, influenza A, and influenza B in a single sample with a single device. Thus, the Q Ag combo test is a promising tool for the detection and differentiation of SARS-CoV-2, influenza A, and influenza B as it is cost-effective, easy to handle, and enables the detection of multiple viruses using a single device with a short turn-around time.

## Figures and Tables

**Figure 1 diagnostics-13-00032-f001:**
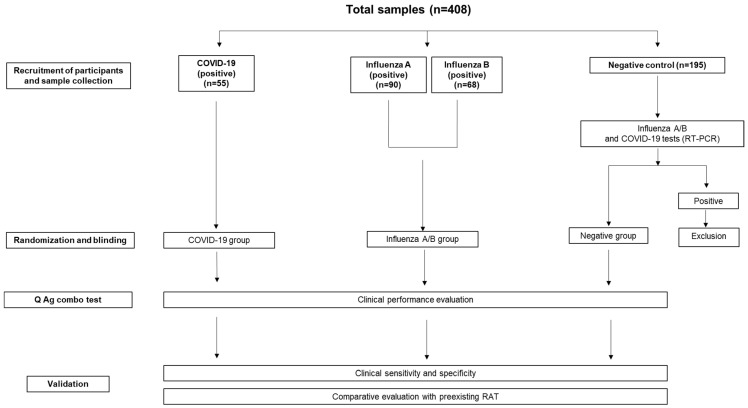
Diagram representing the study design. Following the sample collection, randomization, and blinding, the clinical performance evaluation of the Q Ag combo test was conducted. Subsequently, the Q Ag combo test results were compared with the results from preexisting RAT and with RT-PCR results for validations.

**Figure 2 diagnostics-13-00032-f002:**
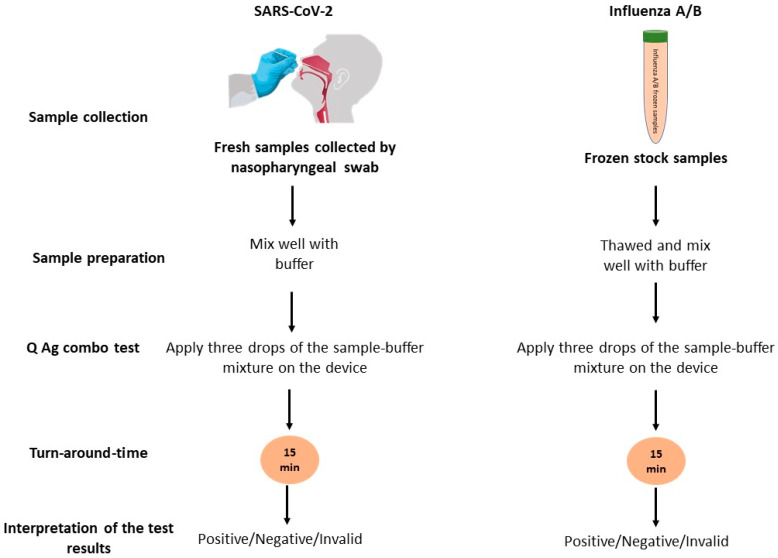
Diagram depicting the test using Q Ag combo. Samples were obtained from participants by nasopharyngeal swab (SARS-CoV-2), or frozen stock (Influenza A/B). Afterwards, the obtained samples were mixed with buffer and about three drops of the mixture were applied to the device. The result can be read and interpreted after 15 min.

**Figure 3 diagnostics-13-00032-f003:**
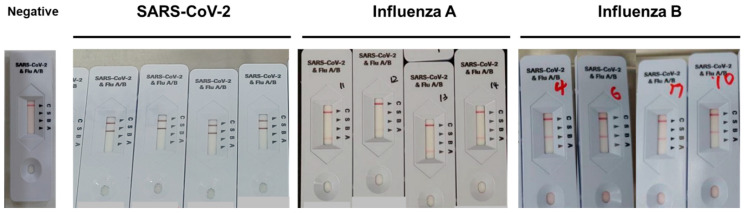
Q Ag combo test results. The control line (C) begins to appear around 3–4 min following the application of the sample-buffer mixture on the device. The other line will also appear next to the test lines when the samples contain antigens of SARS-CoV-2 (S line), influenza A (A-line), or influenza B (B line). One line on the C marker indicates that the test is negative. Two lines, one on C and one on either S, A, or B markers, indicate that the test is positive either for SARS-CoV-2, influenza A or influenza B.

**Table 1 diagnostics-13-00032-t001:** Diagnostic performance of the STANDARD Q COVID/FLU Ag Combo compared to the real-time PCR for detection of SARS-CoV-2.

SARS-CoV-2		PCR Results		Total	Cohen’s Kappa
		Positive	Negative		
STANDARD Q COVID/Flu Ag Combo	Positive	51	1	52	0.940
	Negative	4	194	198	
Total		55	195	250	

Sensitivity, (n), %: (51/55), 92.73% (95% CI: 82.41–97.98%). Specificity, (n), %: (194/195), 99.49% (95% CI: 97.18–99.99%). Positive predicted value, (n), %: (51/52), 98.08% (95% CI:87.82–99.72%). Negative predicted value, (n), %: (194/198), 97.98% (95% CI: 94.97–99.20%). Abbreviation: CI, confidence interval.

**Table 2 diagnostics-13-00032-t002:** Diagnostic performance of the STANDARD Q COVID/FLU Ag Combo compared to the real-time PCR for detection of influenza A.

Influenza A		PCR Results		Total	Cohen’s Kappa
		Positive	Negative		
STANDARD Q COVID/Flu Ag Combo	Positive	83	0	83	0.941
	Negative	7	195	202	
Total		90	195	285	

Sensitivity, (n), %: (83/90), 92.22% (95% CI: 84.63–96.82%). Specificity, (n), %: (195/195), 100% (95% CI: 98.13–100%). Positive predicted value, (n), %: (83/83), 100% (95% CI: 95.65–100%). Negative predicted value, (n), %: (195/202), 96.53% (95% CI: 93.19–98.27%). Abbreviation: CI, confidence interval.

**Table 3 diagnostics-13-00032-t003:** Diagnostic performance of the STANDARD Q COVID/FLU Ag Combo compared to the real-time PCR for detection of influenza B.

Influenza B		PCR Results		Total	Cohen’s Kappa
		Positive	Negative		
STANDARD Q COVID/Flu Ag Combo	Positive	62	1	63	0.928
	Negative	6	194	200	
Total		68	195	263	

Sensitivity, (n), %: (62/68), 91.18% (95% CI: 81.78–96.69%). Specificity, (n), %: (194/195), 99.49% (95% CI: 97.18–99.99%). Positive predicted value, (n), %: (92/63), 98.41% (95% CI: 89.76–99.77%). Negative predicted value, (n), %: (194/200), 97% (95% CI: 93.77–98.58%). Abbreviation: CI, confidence interval.

**Table 4 diagnostics-13-00032-t004:** Sensitivities of the STANDARD Q COVID/FLU Ag Combo for detection of SARS-CoV-2 according to days after symptom onset (DSO) and the Ct value of *RdRp* to the collection of samples.

	Positive	Negative	Sensitivity, %
DSO			
0–3 days	22	0	100% (95% CI, 84.56–100%)
4–6 days	22	0	100% (95% CI, 84.56–100%)
>7 days	7	4	63.64% (95% CI, 30.79–89.07%)
Ct Value of *RdRp*			
≤20	27	2	93.10% (95% CI, 77.23–99.15%)
20 < Ct ≤ 30	16	2	88.89% (95% CI, 65.29–98.62%)
30 < Ct < 40	8	0	100% (95% CI, 63.06–100%)

Abbreviations: DSO, Duration from symptom onset to collection of samples; *RdRp*, RNA-dependent RNA polymerase; Ct, cycle threshold; CI, confidence interval.

**Table 5 diagnostics-13-00032-t005:** Agreement of the STANDARD Q COVID/FLU Ag Combo with the Panbio COVID-19 test kit for detection of SARS-CoV-2, and SD Bioline Ultra Rapid test kit for influenza A and influenza B.

		SARS-CoV-2				Influenza A				Influenza B			
		PANBIO COVID-19		Total	Cohen’s Kappa Value	SD Bioline Influenza		Total	Cohen’s Kappa Value	SD Bioline Influenza		Total	Cohen’s Kappa Value
		Positive	Negative			Positive	Negative			Positive	Negative		
STANDARD Q COVID/FLU Combo Test	Positive	46	6	52	0.924	83	0	83	0.983	14	48	62	0.299
	Negative	0	198	198		2	200	202		1	200	201	
Total		46	204	250		85	200	285		15	248	263	

Abbreviations: CI, confidence interval.

## Data Availability

Data used in this study are available on reasonable request.

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
