# Peer review of "Performance Evaluation of STANDARD Q COVID/FLU Ag Combo for Detection of SARS-CoV-2 and Influenza A/B"

_diagnostics, 2022, doi:10.3390/diagnostics13010032_

Round 1

Reviewer 1 Report

Evaluation of a combo antigen test (COVID+FLU) in comparison with RT-PCR and single antigen tests. The test presents high sensitivity and specificity values that make it useful and efficient for the rapid detection of those multiple targets.

The paper is well written, it is clear and easy to follow and the authors present enough data and solid conclusions. The subject is original, since these combo tests have been launched recently and has high practical application.

Minor comments:

Summary: Last lines establish excellent agreement with pre-existing RAT results, but results presented with Flu B antigen show marked discrepancies.

Introduction: In the first paragraph, the authors refer almost exclusively to the viral causes of respiratory infection. When they mention the most common etiologies, they cite COVID, the flu, pneumococci, and the common cold. I would replace pneumococci with SRV, which hasn't even been mentioned.

Table 3 refers to Influenza B but at the heading, there is Influenza A (mistake?).

At table 4 the authors show incleased sensitivity al low cts (<20) and it decreases with the elevation of cts (20-30) but with extremely high cts (>30) sensitivity is again high. Although they try to justify this strange effect at Discussion, I don’t think they succeed. In my opinion, this can be due to the low number of samples included, especially at this last group, that does’t allow reaching statistical significance.

Again they try to justify, at Discussion, the discrepancies found in Flu B, saying that the samples have undergone freeze-thaw cycles. I wonder why that should affect the single antigenic test but not the combo?

Author Response

The authors would like to thank the Reviewers for their specific and helpful comments on the manuscript. The authors have carefully taken the comments into consideration and have made a revision to the manuscript to address the Reviewers’ concerns. In this revised version, the authors have edited the title, specified the samples that were collected through nasopharyngeal swabs, added information on the source of the influenza A/B frozen-stored samples, edited the figure 1, and 3, and addressed reviewers’ concerns regarding the discussion section.

Detail responses to the Reviewers are reported as follows:

Reviewer 1

Evaluation of a combo antigen test (COVID+FLU) in comparison with RT-PCR and single antigen tests. The test presents high sensitivity and specificity values that make it useful and efficient for the rapid detection of those multiple targets.

The paper is well written, it is clear and easy to follow and the authors present enough data and solid conclusions. The subject is original, since these combo tests have been launched recently and has high practical application.

Minor comments:

  1. Summary: Last lines establish excellent agreement with pre-existing RAT results, but results presented with Flu B antigen show marked discrepancies.

Response: We have corrected this section to “The agreement analysis of the Q Ag combo test with the RT-PCR results demonstrated excellent outcomes, making it useful and efficient for the detection of SARS-CoV-2, influenza A, and influenza B.”

(Index kappa: 0.940, 0.941, and 0.928 for SARS-CoV-2, influenza A, and influenza B, respectively.)

Page: 1, Line: 21-23.

In addition, as comparison with pre-existing RAT is not the main aim of this study in the Results and Conclusion of the abstract, we removed “preexisting RAT” in the first part of the abstract.

Line 14. Before revision: “compared with real-time PCR and pre-existing RAT”

               After revision: “compared with real-time PCR”.

  1. Introduction: In the first paragraph, the authors refer almost exclusively to the viral causes of respiratory infection. When they mention the most common etiologies, they cite COVID, the flu, pneumococci, and the common cold. I would replace pneumococci with SRV, which hasn't even been mentioned.

Response: As suggested by the reviewer, we have replaced the pneumococci with RSV infection. We also specify it into four “viral” RTIs.

Page: 1, Lines: 31-32

“Among the numerous RTIs that can be named, there are four main viral RTIs that are commonly found nowadays: COVID-19, influenza, respiratory syncytial virus (RSV) infection, and the common cold”

  1. Table 3 refers to Influenza B but at the heading, there is Influenza A (mistake?).

Response: We have corrected the heading of table 3.

Page: 7, Table: 3

  1. At table 4 the authors show incleased sensitivity al low cts (<20) and it decreases with the elevation of cts (20-30) but with extremely high cts (>30) sensitivity is again high. Although they try to justify this strange effect at Discussion, I don’t think they succeed. In my opinion, this can be due to the low number of samples included, especially at this last group, that does’t allow reaching statistical significance.

Response: We have added the statement about the small sample size in (30<Ct<40) as a limitation for the determination of the results.

Page: 9, Lines: 304-306

“However, we also considered the small number of samples included in this group (30<Ct<40) as the limitation that makes it difficult to determine whether the outcome of this group is a true finding, given that a small sample size may reduce the statistical power. “

  1. Again they try to justify, at Discussion, the discrepancies found in Flu B, saying that the samples have undergone freeze-thaw cycles. I wonder why that should affect the single antigenic test but not the combo?

Response: In our study, we found that among the samples that have been confirmed positive for influenza B by RT-PCR, most of them also result in positive outcomes when being assessed using the Q Ag combo test, only a few also give a positive result when being assessed using SD Bioline Influenza Ultra Rapid test kit (preexisting RAT). Previously, a manufacturer-independent evaluation of the Ag-rapid detection kit’s (Ag-RDT) limit of detection reported that 11 among 19 Ag-RDT showed decreased sensitivity up to 20 folds when used to assess frozen-stock samples (-80oC). Most of the RATs are intended to be point-of-care tests, hence requiring fresh samples to meet the highest sensitivity and specificity as stated by the manufacturer. Therefore, we assumed that the usage of frozen storage influenza B samples affected the sensitivity of the preexisting RAT, but not the Q Ag combo test, resulting in a low Cohen’s Kappa value for agreement between the two test kits.

Page: 9, Lines: 317-335

Reviewer 2 Report

This is a very excellent study on the performance of combination antigen rapid test kit.

My major comment: This study is not totally field study as the researcher used known positive frozen sample for influenza A and B. I suggest to remove “from the Patients with Suspected Respiratory Infections”.

Here my minor comments for the improvement of the manuscript:

1.     Line 32: The reference #3 has been deleted from the online database. Need to find other reference to replace.

2.     Line 74: The word participants is not appropriate because some of the samples were taken from storage. I suggest to use the word ‘samples’ or ‘specimens’.

3.     Line 85: May need to declare the source of samples.

4.     Figure 1: In flowchart, I suggest only one control arm to avoid confusion. The control box beside COVID-19 (Positive) can be remove and the control box beside Influenza B (positive) can be maintained with text “COVID-19 and Influenza A/B (Negative)”. This will make total study samples as 408 (NOT 250 and 353 as currently stated).

5.     Line 96-102: Sample collection should be specific for COVID-19 positive and Control

6.     Line 118-122: In this paragraph, author should state how many observers of the cassette and are they were blinded. If only one observer and he is not blinded, please state single unblinded observer.

7.     Figure 3: I have difficulty to se positivity of cassette #12 and #22.

8.     In the discussion, please add

a.     Limitation of this study includes samples for influenza A/B were from -80 storage and single unblinded reader.

b.     Low Cohen’s Kappa value for agreement of influenza B antigen test kit. Please find the references for the specificity of the comparative test kit.

9.     Line 334: Please state somewhere in the methodology why Informed Consent Statement is Not applicable for the specimen taking of COVID-19 and control participants.

Author Response

The authors would like to thank the Reviewers for their specific and helpful comments on the manuscript. The authors have carefully taken the comments into consideration and have made a revision to the manuscript to address the Reviewers’ concerns. In this revised version, the authors have edited the title, specified the samples that were collected through nasopharyngeal swabs, added information on the source of the influenza A/B frozen-stored samples, edited the figure 1, and 3, and addressed reviewers’ concerns regarding the discussion section.

Detail responses to the Reviewers are reported as follows:

Reviewer 2

This is a very excellent study on the performance of a combination antigen rapid test kit.

  1. My major comment: This study is not totally field study as the researcher used known positive frozen sample for influenza A and B. I suggest to remove “from the Patients with Suspected Respiratory Infections”.

Response: As suggested by the reviewer we have altered the title by removing “from the patients with suspected respiratory infection” from the title.

Page 1 (Title: Performance Evaluation of STANDARD Q COVID/FLU Ag Combo for Detection of SARS-CoV-2 and Influenza A/B)

Here my minor comments for the improvement of the manuscript:

  1. Line 32: The reference #3 has been deleted from the online database. Need to find other reference to replace.

Response: We have added a new reference to replace the previous one as suggested by the reviewer.

Page 10, line: 372

(Tregoning, J.S.; Schwarze, J.r. Respiratory viral infections in infants: causes, clinical symptoms, virology, and immunology. Clin. Microbiol. Rev. 2010, 23, 74-98.)

  1. Line 74: The word participants is not appropriate because some of the samples were taken from storage. I suggest to use the word ‘samples’ or ‘specimens’.

Response: As suggested by the reviewer we have replaced the word “participants” with “samples”

Page: 2, Lines:74-78

“A total of 408 samples were evaluated in this study. All the samples were assessed by real-time PCR for detection of the SARS-CoV-2, influenza A, and influenza B viruses.”

  1. Line 85: May need to declare the source of samples.

Response: We have added the information regarding the source of the samples

Page: 2, Lines: 83-86

“All positive samples of influenza B and a portion of the influenza A samples were purchased from an overseas biobank (National Institute of Hygiene and Epidemiology, Hanoi, Vietnam; and Instituto de Investigación Nutricional, Lima, Peru).”

  1. Figure 1: In flowchart, I suggest only one control arm to avoid confusion. The control box beside COVID-19 (Positive) can be remove and the control box beside Influenza B (positive) can be maintained with text “COVID-19 and Influenza A/B (Negative)”. This will make total study samples as 408 (NOT 250 and 353 as currently stated).

Response: As suggested by the reviewer we have removed the redundancy of the “negative control box” in the flowchart.

Page: 3, Figure 1.

  1. Line 96-102: Sample collection should be specific for COVID-19 positive and Control

Response: As suggested, we have specified the samples that were collected by nasopharyngeal swab.

Page: 3, Line: 98

“The nasopharyngeal samples for assessment of COVID-19 and negative control were collected as described previously”

  1. Line 118-122: In this paragraph, author should state how many observers of the cassette and are they were blinded. If only one observer and he is not blinded, please state single unblinded observer.

Response: As suggested, we added the statement of the number of technicians who observed the cassette.

Page: 4, Line: 120.

“The test was conducted by three technicians who were blinded to avoid any bias from the observer.”

  1. Figure 3: I have difficulty to se positivity of cassette #12 and #22.

Response: We edited the figure by replacing cassettes #12 and #22 with other cassettes for influenza B assessment with vivid positive results.

Page: 4, Figure 3.

  1. In the discussion, please add:

a. Limitation of this study includes samples for influenza A/B were from -80 storage and single unblinded reader.

Response: As suggested by the reviewer, we added the storage sample as a limitation in our study.

Page: 10, lines: 336-345

“The use of frozen-stored samples which leads to the discrepancy of the preexisting RAT and Q Ag combo test for the detection of influenza B may become the biggest limitation in our study. The frozen-stored samples for the influenza A/B test were used in this study due to our inability to recruit participants with influenza A/B infections. As reported before, during the second winter season of the COVID-19 era (2021-2022 winter season), there was no influenza outbreak in South Korea [31]. Nevertheless, the Q Ag combo tests demonstrated a high sensitivity when used for the assessment of freshly-collected or frozen-stored samples, making it a promising POC test kit with high performance for the detection of SARS-CoV-2, influenza A, and influenza B with only a single device, regardless of the sample condition.”

As the observer, as stated in the methodology, there are three technicians that conducted the assessment and observed the cassettes, hence we exclude it from the limitation of the study.

Page: 4, Line: 120.

“The test was conducted by three technicians who were blinded to avoid any bias from the observer.”

b. Low Cohen’s Kappa value for agreement of influenza B antigen test kit. Please find the references for the specificity of the comparative test kit.

Response: As suggested, we have added a reference for the specificity of the comparative test kit.

Pages: 9-10, lines: 322-335

“The SD Bioline Influenza Ultra test kit (comparative test kit/preexisting RAT) manufacturer claimed that the sensitivity and specificity of the kit are up to 91.5% and 98.7% for samples from the nasopharyngeal swab, and up to 91.7% and 98.9% for samples from the nasopharyngeal aspirate. However, there is no information about the sensitivity and specificity of the SD Bioline Influenza Ultra-test kit for assessment of the samples that have undergone freeze-thawed cycles. Note that the influenza B samples for our assessment were frozen stock samples. Previously, a manufacturer-independent evaluation of the Ag-rapid detection kit’s (Ag-RDT) limit of detection reported that 11 among 19 Ag-RDT showed decreased sensitivity up to 20 folds when used to assess frozen-stock samples (-80oC) [30]. Most of the RATs are intended to be point-of-care tests, hence requiring fresh samples to meet the highest sensitivity and specificity as stated by the manufacturer. Therefore, we assumed that repeated freeze–thawed cycles affected the sensitivity of the SD Bioline Influenza Ultra test kit (preexisting RAT), but not the Q Ag combo test kit, resulting in a low Cohen’s Kappa value for agreement between the two test kits. “

  1. Line 334: Please state somewhere in the methodology why Informed Consent Statement is Not applicable for the specimen taking of COVID-19 and control participants.

Response: We have corrected the information about the “Informed Consent Statement”.

Page: 10, line: 362-363

Informed Consent Statement: Individuals who participated agreed to this study and submitted their wri

Round 2

Reviewer 2 Report

Can accept in present form.